# Alzheimer’s Disease and Tau Self-Assembly: In the Search of the Missing Link

**DOI:** 10.3390/ijms23084192

**Published:** 2022-04-10

**Authors:** Andrea González, Sandeep Kumar Singh, Macarena Churruca, Ricardo B. Maccioni

**Affiliations:** 1Laboratory of Neurosciences and Functional Medicine, International Center for Biomedicine (ICC) and Faculty of Sciences, University of Chile, Santiago 7800020, Chile; azgonzal@uc.cl (A.G.); macarenachurruca@gmail.com (M.C.); 2Indian Scientific Education and Technology Foundation, Lucknow 226002, India; sandeeps.bhu@gmail.com; 3Department of Neurology, Faculty of Medicine, University of Chile, Santiago 7500922, Chile

**Keywords:** Alzheimer’s disease, tau protein, polyanions, polyphosphates, neurofibrillary tangles

## Abstract

Alzheimer’s disease (AD) is a multifactorial neurodegenerative disease characterized by progressive cognitive impairment, apathy, and neuropsychiatric disorders. Two main pathological hallmarks have been described: neurofibrillary tangles, consisting of tau oligomers (hyperphosphorylated tau) and Aβ plaques. The influence of protein kinases and phosphatases on the hyperphosphorylation of tau is already known. Hyperphosphorylated tau undergoes conformational changes that promote its self-assembly. However, the process involving these mechanisms is yet to be elucidated. In vitro recombinant tau can be aggregated by the action of polyanions, such as heparin, arachidonic acid, and more recently, the action of polyphosphates. However, how that process occurs in vivo is yet to be understood. In this review, searching the most accurate and updated literature on the matter, we focus on the precise molecular events linking tau modifications, its misfolding and the initiation of its pathological self-assembly. Among these, we can identify challenges regarding tau phosphorylation, the link between tau heteroarylations and the onset of its self-assembly, as well as the possible metabolic pathways involving natural polyphosphates, that may play a role in tau self-assembly.

## 1. Introduction: Alzheimer’s Pathogenesis

Alzheimer’s disease (AD) is a multifactorial, neurodegenerative disease, which is characterized by progressive cognitive decline and behavioral disorders of patients. Within its prognosis, three clinical stages are recognized: (1) The hippocampal stage, where the first signs of memory and judgment or executive capacity defects are evident. (2) The parieto-temporo-occipital stage (also prefrontal), associated with moderate and more severe dementia, with language disorders and others. (3) Stage of global damage, the patient presents loss of language, dysphagia, rigidity, and prostration. Aging is one of the main risk factors for AD since most of the cases are related to elderly individuals, older than 65 years, affected by so-called “sporadic AD”, currently the most common type of dementia in the senile population [1,2].

Nowadays, 50 million people are affected with AD worldwide, and this number is estimated to grow to 82 million by 2030 and 152 million by 2050 (World Alzheimer’s Report, 2018). In the global context, the costs of dementia were USD 1 trillion in 2018; 25% of this is direct medical costs and 75% indirect social costs for caregivers. In this context, this epidemic disorder is concerning for public health, with efforts focused on its prevention and treatment.

It should be noted that one of the major challenges to developing an innovative therapy or an effective diagnostic tool for AD is precisely the fact that it is a multifactorial disease. A risk factor associated with the origin and evolution of AD is the alteration of glucose metabolism that causes insulin resistance and type II diabetes mellitus (DM-II). This is demonstrated by several epidemiological studies [3,4,5]. For example, deterioration in glucose metabolism can decrease the energy available for cognitive information processes and lead to synaptic dysfunction. This happens in patients with mild cognitive impairment. If insulin resistance is maintained for longer, mitochondrial damage may be greater and oxidative damage may increase, leading to neurodegeneration in the brain. However, the mechanisms of how insulin resistance and DM work as risk factors for AD are still uncertain [6]. Another factor related to AD is depression; there is comorbidity in 50% of cases of older people with AD. Depression is considered part of the initial symptoms or a preclinical phase [7]. A relationship between high levels of beta-amyloid with increased depressive and anxiety symptoms has been evidenced. Supporting this hypothesis, neuropsychiatric symptoms could be predictors of AD [8,9,10]. This association is relevant because several diseases resulting from physiological alteration of depression (deregulation of the hypothalamic–pituitary axis, neuroinflammation, and cerebrovascular changes) have been associated with amyloid deposition [11]. Stress events may lead to depression [12] altering the HPA axis by increasing glucocorticoid production [13]. The “damage, signal” caused by stress results in the activation of the defense system mediated by glial cells of the innate immune system [14,15]. This has been related to previous stages of AD [16] and concurs with our theory of neuroimmunomodulation [11,17].

AD is characterized by two major hallmarks: (i) senile plaques (SP), composed of deposits of the amyloid-β (Aβ) peptide of 39 to 42 aminoacidic residues, generated by the proteolytic excision of the amyloid precursor protein (APP) by β and γ secretases, in the extracellular space [18]; and (ii) neurofibrillary tangles (NFT), derived from the progressive aggregation of hyperphosphorylated tau protein, inside neurons, that derive from tau assembly into oligomeric structures named “paired helical filaments” (PHF). PHFs block transport systems in neurons, thus affecting synaptic transmission [19]. The abnormal hyperphosphorylation of tau has deleterious pathological effects which are directly involved in neurodegeneration. In this context, tau protein, with its extensive pathological role in neurodegenerative diseases, is a very a promising target for therapeutic interventions. Additionally, the previously mentioned amyloid beta-protein has been the main therapeutic target in a vast array of research and treatments in the past 20 years. Nevertheless, the results of clinical and preclinical trials have been inconclusive, showing the need to find a new treatment strategy for AD [20]. Donepezil is one of the few drugs approved for AD treatment, as it is a specific inhibitor of the AchE, and recently its high affinity for human transferrin [21] has been observed, indicating an involvement in iron homeostasis.

After clarifying the major role of tau hyperphosphorylations in AD pathogenesis, a major question was to elucidate the molecular signals “upstream” that lead to the pathological phospho-tau variant. AD is a summatory of neuroinflammatory events that result from the innate immunity phenomenon of activation of microglial cells by the so-called “damage signals” that include a large set of molecular or physical factors [22,23] (Figure 1). The neuroinflammatory response involves an over-release of proinflammatory cytokines that finally signal neuronal receptors, resulting in the formation of a stable CDK5/p25 complex responsible for part of tau hyperphosphorylations. This is the basis of the **Neuroimmunomodulation theory of Alzheimers disease** [1,14,17,22,23]. In essence, AD is a consequence of disturbances in the cross-talks between microglial and neuronal cells that finally renders a modified phospho-tau variant followed by misfolding of tau’s structure. Thus, the neuroimmunomodulation theory has been able to explain the mechanism underlying the molecular basis of AD, allowing us to appropriately select the targets of future therapies, as well as early biomarkers for this disease. In the neuroimmunomodulation theory, the “damage signals” (such as Aβ peptide, reactive oxygen species, iron overload, vitamin B12 deficiency, free tau fragments, etc.) activate the microglia, which switch from a quiescent state to an activated (M1) state. The M1 microglia generates proinflammatory cytokines, such as IL-6 and TNF-α, generating a proinflammatory microenvironment in the brain. These cytokines also promote the assembly of the CDK5/p25 complex, promoting the hyperphosphorylation of tau. The latter eventually leads to neurodegeneration, in which some PHF fragments can act as novel danger signals in surrounding microglia, in a cyclic event.

## 2. Role of Glial Cells and Immune Response in Tauopathies

Glial cells play an important role in maintaining neuronal function and survival [24] and tau can affect this functionality. Tau-expressing glial cells have been shown to wield toxicity via both cell-autonomous effects in glial cells and non-autonomous alterations in neurons. Neuroinflammation has been studied and reported in brain tissues from different tauopathies. Observations from the data obtained from genetic studies have acknowledged the glial compartment as the primary moderator of this typical process. Both astrocytes and microglia synergistically contribute to inflammation in the central nervous system in the case of tauopathies. Neuroinflammation’s impact on tau pathology is based on the timing and magnitude of the response from the immune system. Microglial cells are a class of the innate immune system and are characterized as dweller macrophages of the CNS. They can engulf different extracellular substances such as apoptotic cells, cell debris, and microbes. Phagocytic activity of the microglial cell is an important player and imparts a significant role in physiological processes including adult neurogenesis, CNS development, pathogen defense, and synaptic homeostasis.

Activation of microglial cells is also a very common and known feature of aging and neurodegenerative diseases [25]. In the cellular milieu in tauopathies, reactive microglial cells are found around NFTs in the AD brain as well as in supranuclear palsy, Parkinson’s disease and Lewy bodies disease [26,27]. The main function could be the removal of misfolded tau at the pathological sites. Microglia can capture and engulf the tau aggregates under in vitro and in vivo conditions, and this whole process depends on signaling pathways of CX3C chemokine receptor 1 [28]. Misfolded tau protein acts like a stimulus to activate the microglia by its interaction [29]. During the initial stage of tau pathogenesis, there is production of various anti-inflammatory interleukins such as IL-10, IL-4, IL-13, and transforming growth factor β (M2 phenotype), which can control the local immune response and encourage tissue repair. Nevertheless, as the disease progresses slowly, the microglia remain activated, resulting in the change from M2 to M1 phenotype, ultimately aggravating tauopathy through the embellished release of reactive oxygen species (ROS), nitric oxide (NO), and proinflammatory cytokines such as the interleukins IL-1β, IL-6, IL-12, IL-23, and also tumor necrosis factor α (TNF-α) [30].

Other glial cells called astrocytes are also very important from a tauopathy perspective. Astrocytes are the most common and abundant cells and are ubiquitous in all regions of the CNS. Astrocytes are important players and have a role in various functions, including those that are metabolic and physical, and provide support for detoxification neurons, cell migration guidance, and regulation of metabolism related to energy. Under disease conditions or any sort of injury, astrocytes undergo chronic activation called astrogliosis [31]. In the same way as microglia, the reactive astrocytes can obtain the proinflammatory A1 phenotype or A2 status defined by upregulated neuroprotective molecules such as vascular endothelial growth factor, brain-derived neurotrophic factor and basic growth factors, which can promote synaptic repair and neuronal survival [32]. The AD brain has a higher prevalence of large A1 astrocytes. This type of inequality in the ratio of A1/A2 is directly caused by microglia dysregulation. IL-1α, TNF-α, and C1q-secreted b-activated microglia are essential and quite enough to polarize astrocytes toward the A1 phenotype [33,34].

## 3. Hyperphosphorylation of Tau: The Influence of Protein Kinases and Phosphatases during the Course of AD

Tau is a microtubule-associated protein (MAP) that is probably involved in the control of axonal transport [35]. The gene encoding tau is located in chromosome 17, region 17q 21, in humans [36]. It is structurally considered an elastic molecule and no sequential structure has been detected [37]. Its plasticity allows some of its exons (2,3,4A, 6, 8,10 and 14) to be spliced in an alternative way. This generates 6 isoforms, and it contains two major domains, an amino terminal domain and a carboxy-terminal domain. The first one is denominated ‘projection domain’, which is rich in proline and also has an acid region. The C-terminal is the principal binding domain of the microtubules and is organized by three (3R) or four (4R) internal repetitions [38].

The two main functions of tau are, under normal conditions, to provide stability to the microtubules (MT) and to articulate the transport system of signaling molecules and cellular components [20,39]. Those functions are disrupted once tau is hyperphosphorylated during the course of AD [40]. This process increases the number of binding sites in the same phosphorylated tau molecule [41]. Hyperphosphorylated tau captures native tau and other microtubule-associated proteins, causing the disassembly of microtubules [19,42].

Subsequently, there is a destabilization of the cytoskeleton, produced by the alteration of tau-dependent cellular functions, such as vesicular and organelle transport, axonal growth and nerve signal propagation [43]. This anomaly is known as tauopathy and is present in many neurodegenerative diseases [39]. There are 20 diseases categorized as tauopathies, which are sub-divided into two groups; Alzheimer Disease is part of the secondary group and it is also the most preponderant. The secondary group is characterized for the presence of both an intracellular tau pathology and an extracellular amyloid plaque deposit. The particularity of these tauopathies is the formation of insoluble deposits called neurofibrillary tangles (NFT) in the three (3R) and four (4R) isoforms. The dysfunctionality caused by NFT is manifested from the soma to the dendrites, and the most commonly affected regions of the brain are the entorhinal cortex, the hippocampus and neocortex [44].

Studies have demonstrated how an increased activity of kinases, such as CDK5, and downregulation of phosphatases, influences tau hyperphosphorylation, leading to the oligomer formation of tau [45,46,47]. The deregulation of CDK5 is due to the formation of the CDK5/p25 complex, a product of p35 splitting, possibly as a result of oxidative stress and amyloid peptides to which the neuron has been exposed. The conversion from p35 to p25 occurs through proteolysis. The result is p25, a fragment of the protein that is neurotoxic and has an active and a totally extended conformation. The conformational change suffered by p35 produces p25. Additionally, that conformational change impacts the way CDK5 activates, given the fact that the latter activation lasts longer than p35. This conversion results in CDK5 hyperactivity, and subsequently, a possible hyperphosphorylation of tau protein and neurofilaments, along with a cytoskeletal alteration and eventually neuronal death [48]. 

Their conformational structure changes from an α-helix to a β-sheet structure, which facilitates the formation of the oligomers. Thus, phosphorylation of tau is the main post-translational modification that is involved with AD [41].

In AD, the nature of the tau amino acids that are the target of the protein kinases is just as important as the proper hyperphosphorylation. Thus, the key amino acid phosphorylations are those critical for tau conformational changes, from an alpha-helix to a beta-sheet structure. In that regard, it was demonstrated in vitro that oxidative stress promotes tau dephosphorylation at the Tau1 epitope in SHSY5Y cells [49]. The latter was dependent on the activity of the cdk5/p35 complex, since an increase in the substrate phosphorylation as well as the complex association were observed. Additionally, oxidative stress induced a decrease in the amount of inhibitor-2 bound to phosphatase PP1, associated with an increased phosphorylation of the inhibitor-2 protein. Thus, hyperphosphorylation of tau relies on a shift of the balance between the kinases and phosphatases, in which the upregulation of the kinases’ activity exceeds the phosphatase activity.

However, there are other post-translational modifications of tau protein, besides hyperphosphorylation, that play different roles during the pathological processes leading to AD. These modifications include truncation and glycosylation; both of which occur in early stages of AD. In a study by Takahashi et al. (1999), it was observed that tau proteins were abnormally glycosylated in the brain of AD patients, which was not detected in control patients [50]. Other types of post-translational modifications are glycation, nitration, ubiquitination and polyamination [39].

Liu et al. (2002) have shown that abnormal in vitro glycosylations modulate the phosphorylation of tau by the kinases PKA, GSK-3 and CDK-5 [51], which in turn, inhibit dephosphorylation by the phosphatases PP2A and PP5 [52]. Interaction between many post-translational modifications may be necessary to induce the oligomer tau formation [41].

Additionally, the role of the microtubule regulating protein kinase MARK4 in AD has been demonstrated, as its overexpression increases tau hyperphosphorylation. In that regard, this kinase is inhibited by AChE inhibitors Donepezil and Rivastigmine [53]. Similarly, irisin, a hormone usually increased during physical exercise, has also an inhibitory potential over MARK4 [54]

Another aspect to be considered is diagnosis, as currently in AD there are several radiotracers used for positron-emission tomography (PET) that bind tau filaments, such as THK5317, THK5351, AV-1451, and PBB3 [55]. APN-1607 is one of the newest radiotracers successfully bound to PHF [56]. This is important since the design of the tracers depends on the tau structure. Specific phosphorylated tau tracers for PET scans would make the diagnosis more accurate, but their design is still a challenge.

## 4. In Vitro Induction of Tau Aggregates: Polyanions and Polyphosphates

To further investigate the mechanism of tau aggregation, a recombinant tau protein (rTau) was obtained. Contrary to what was observed with the Aβ-peptide [57] and α-synuclein [58], the rTau protein did not oligomerize spontaneously when incubated at physiological pH, temperature, ionic strength, and concentration (1–10 μM) [59]. This process of in vitro tau polymerization required inducer molecules such as heparan sulfate, RNA, heparin and arachidonic acid in addition to a kinase to catalyze the reaction [60,61,62]. It should be noted that the above-mentioned catalyzers can act as polyanions due to its negatively charged groups (e.g., sulfate). In vitro, polyanions enhance tau aggregation by stabilizing protein domains that are key for this process. Hyperphosphorylated tau, which is not attached to the microtubules, binds in pairs, forming paired helical filaments (PHF). The resemblance between the binding sites of microtubule and polyanions confirm the hypothesis that microtubule stabilization prevents PHF formation by blocking tau-polyanion interaction sites [63].

When full-length tau polymerization was compared with different catalyzers, it was observed that the process occurred faster with arachidonic acid than with heparin at physiological concentrations [64]. This could be because arachidonic acid acts in micellar form and promotes fibrillization by binding tau on its anionic surfaces, and proceeds via a partially folded thioflavin S (ThS)-positive intermediate, which is time-dependent but does not display a lag phase [65,66,67].

In heparin-induced tau polymerization, contrary to what was observed with arachidonic acid, there is a pronounced lag phase [64,68]. Only the addition of nucleation seeds can reduce it. Since heparin has a polyanionic nature (several sulfate groups), the highly negative charges may provide a substrate that resembles the surface of the MT’s [69,70]. It should be noted that none of the heparin molecules assemble with tau during its polymerization, thus it is plausible that heparin plays a kinetic rather than a structural role [70]. Heparin catalyzes tau aggregation involving two different types of dimers (cysteine-dependent or cysteine-independent dimers) and higher-weight oligomers [71]. These structures can aggregate into intermediate filaments in a solution called granular tau [72]. The first step of dimer or oligomer formation is the phosphorylation of tau at key protein residues. After the phosphorylation, conformational changes and truncation of tau by caspases occurs. After phosphorylation, massive conformational changes and caspase truncation of tau occurs, which appears to be driven by proteolytic truncation at both ends of the molecule [73]. In vitro experiments reveal that the C-terminal moiety of tau inhibits its aggregation [74]. However, all the above-mentioned catalyzers do not possess a phosphate group, which is the main substrate of the kinases and is responsible for the phosphorylation.

Single-molecule fluorescence was used to study the conformational steps that occur during the initiation of tau oligomers’ formation [75]. Authors have found that another polyanion, polyphosphate, triggers tau oligomerization due to electrostatic interactions of the polyanion and the cationic domains on tau. The mechanisms point to an intramolecular scaffold that promotes tau conformations susceptible to self-aggregation. On the other hand, intermolecular interactions may increase a local tau environment. Recently, it has been demonstrated that polyphosphates (polyP) can induce tau aggregation in vitro. Wickramasinghe et al. (2019) described two mechanisms relevant to the initiation of tau aggregation in the presence of cytoplasmic polyP: the first is changes in the conformational ensemble of monomer tau, and the second is noncovalent cross-linking of multiple tau monomers [76]. These conformational changes throughout full-length tau lead to the diminishment of long-range interactions between the termini coupled with compaction of the microtubule-binding and proline-rich regions. Even though the proline-rich and microtubule-binding regions both contain poly-P binding sites, the proline-rich region is pivotal for compaction of the microtubule-binding region upon binding. It was also observed that both the magnitude of the conformational change and the aggregation of tau are dependent on the chain length of the polyP polymer. Longer polyP chains are more effective in intermolecular, noncovalent cross-linking of tau. These observations provide an understanding of the initial steps of tau aggregation through interaction with a physiologically relevant aggregation inducer.

## 5. In Vivo Tau Polymerization: Unveiling the Black Box

All the above catalyzers were studied in vitro, but are these molecules physiologically relevant? Under normal conditions, levels of free cytoplasmic arachidonic acid are quite low, but under certain circumstances, such as ischemia, the levels of cytoplasmic free fatty acids can become highly elevated. Increasing concentrations of AA causes negative effects on the function and structure of the neuron cells, for example, mitochondrial inflammation, deregulation in the production of ATP and activation of the nuclear transcription factor kappa (NF-κΒ). Consequently, it is possible that an “arachidonic acid cascade” is initiated by the accumulation of free fatty acids. The activation of this process triggers an increase in the production of reactive oxygen species (ROS). AA is regulated by phospholipase_2_ to maintain its basal levels and perform normal brain function, but the increased enzymatic activity of PLA_2_ causes an excessive release of proinflammatory mediators [77]. In particular, in AD there is an upregulation of cytoplasmic phospholipase2 activity, thus increasing the levels of fatty acids and their metabolites. Glycosaminoglycans (sGAG) (e.g., heparan sulfate) showed an association with the NFT pathology in AD [78,79]. Sulfated glycosaminoglycans generally exist in association with proteins; they do not usually exist in free form. Heparin (HS) is a hyper-sulfated form of heparan sulfate, and it is also sGAG subtype [80]. It has been observed in vitro that sGAG facilitates the polymerization of tau and has a role in determining the helicity of the PHF [81]. It has also been evidenced that it increases the capacity of self-assembly of mutated tau proteins by the presence of sGAG, compared to a native tau protein. Using an antibody for HS, a strong association was observed between HS and the neurofibrillary tangles of AD present in the intracellular and extracellular environment. This is because it reacts specifically with some of the PHFs [82].

However, the latter does not explain the missing substrate, the phosphate. Several hallmarks point out that mitochondrial damage, oxidative stress and impaired glucose metabolism are pivotal in AD for the hyperphosphorylation of tau, and consequently for the generation of tau oligomers [6,83,84,85]. A direct relationship between insulin depletion, which decreases glucose utilization, and increased mitochondrial dysfunction, has been shown to be one of the major changes observed in AD. Mitochondrial damage is related to synaptic impairment associated with increased oxidative stress in the brain [6]. In oxidative damage, an imbalance is observed between the production of ROS and the levels of antioxidants (AO). When the production of ROS in the mitochondria increases, it may exceed the capacity of the AO system, leading to a period of oxidative stress [86].

If this is so, damaged mitochondria may release naturally occurring polyP, such as adenosine-tri phosphate (ATP). Thus, it is plausible that once the concentration in the cytoplasm increases due to the mitochondrial damage, they act as a source for the kinases mentioned above, upregulating their activity and inducing tau aggregation intracellularly.

We propose that the ignition of these signals conveys glucose metabolism impairment, which increases oxidative stress and mitochondrial damage. The latter elevate naturally occurring polyphosphates, and consequently, hyperactivation of kinases such as CDK5 and GSK3β combined with downregulation of phosphatases such as PP2A and PP5. Moreover, this source of polyanions could be responsible for the onset of tau oligomerization and further aggregation into PHFs.

The latter is supported in a work by Joshi et al. (2019), which demonstrates that fragmented mitochondria are released from the microglia and propagate the inflammatory cascade and consequent neurodegeneration [87].

Additionally, it should be considered that ATP has a major role in the control of neuronal activity through microglia-driven negative feedback. It was demonstrated that microglia suppress the neuronal activation in a specific region through their capacity to sense and catabolize extracellular ATP, which is released upon neuronal activation by astrocytes and neurons [88]. ATP recruits microglial protrusions and is converted into AMP by the microglial hydrolyzing enzyme CD39. AMP is then converted into adenosine by CD73, which is expressed in microglia and other brain cells. This nucleoside is the key to the suppression of neuronal activity, via the adenosine receptor A present in neurons, thus establishing the microglia-mediated negative feedback mechanism.

All the above highlights the importance of polyphosphates, in particular of ATP, in the regulation of neuronal activity. If by mitochondrial damage excess ATP is released, it is plausible that upregulation of the cdk5/p25 complex occurs, as well as upregulation of the suppression of neuronal activity that would eventually lead to neurodegeneration.

## 6. Tau Auto-Aggregation and the Hypothesis of Expansion

In AD, a minimum of 30 serine/threonine residues of tau are phosphorylated, and the level of phosphorylation is correlated with the severity of the pathology. It has been demonstrated that purified phospho-tau is capable of self-polymerization in vitro without inducer molecules. Additionally, hyperphosphorylated tau can sequester tau protein from microtubules. This highlights the critical **role of phosphorylation’s in tau oligomerization, and provides evidence that points to this event as one of the first milestones in AD** [89]. In addition, if NFTs appear in the entorhinal cortex in the early stages of AD, and are later found in adjacent areas that are anatomically connected [90], it would mean that NFTs could be transmitted from one neuron to another [91]. So, the expansion from intracellular to extracellular is the next step. Can the PHF from the extracellular region access a nearby neuronal cell? The infectious hypothesis states that extracellular aggregates can access another cell, where they act as nucleation seeds to form larger aggregates of tau.

The propagation of tau from neuron-to-neuron has been proven by experiments with PHFs both in vivo and in vitro [20]. In vivo evidence of this hypothesis has been shown, as it has been demonstrated that extracellular aggregates can induce the formation of these structures inside the cell. The latter is consistent with other studies that show the capacity for transmission of PHFs-like tau aggregation from a murine model of tau pathology (P301S) to a murine model that expresses human tau [92]. Considering that tau is an intracellular protein, we propose that once the initial event is triggered (e.g., glucose metabolism impairment), several signals lead to mitochondrial dysfunction. This, in turn, leads to an increase in naturally occurring polyP concentration (e.g., ATP, GTP), which upregulates the activity of kinases, such as CDK5, hyperphosphorylation of tau and initiation of the self-assembly process. The formation of larger tau aggregates, together with other apoptotic signals (e.g., upregulation of suppression of neuronal activity by microglia), lead to neurodegeneration (Figure 1). Tau aggregates, in particular PHF, can be released to the extracellular once the cell enters apoptosis, where they can access a nearby cell and act as nucleation seeds for the formation of larger tau aggregates, in a cyclic process (Figure 2).

## 7. Conclusions

In the present manuscript, we show some evidence of the influence of PolyP in tau self-assembly in AD and analyze recent literature in regards of the etiology of one of the hallmarks of AD: neurofibrillary tangles and hyperphosphorylated tau. The major questions in this regard are: Where does the source of the phosphate come from? How can polyP be a key element in tau self-assembly? Major metabolic disturbance, closely related to mitochondrial dysfunction, may lead to an increase in naturally occurring polyP (ATP, GTP) in the cytosol, and may upregulate the activity of the kinases or decrease the activity of phosphatases. After carefully considering the revised literature, we hypothesize that ATP, after release to cytosol, may act as a upregulator/substrate of the CDk5 kinase, which, in turn hyperphosphorylates tau. After that, the conformational change in tau structure to β-sheets takes place, leading to the formation of PHF, and finally, to NFT. However, the hypothesis still needs to be verified and the precise molecular mechanisms are still to be elucidated.

## 8. Materials and Methods

The revised literature search was performed in Pubmed Central. Publications were selected based on (i) a search by keywords (e.g., tau, polyphosphates, Alzheimer disease); (ii) impact of journal and (iii) year of publication, with preference given to recent literature.

## Figures and Tables

**Figure 1 ijms-23-04192-f001:**
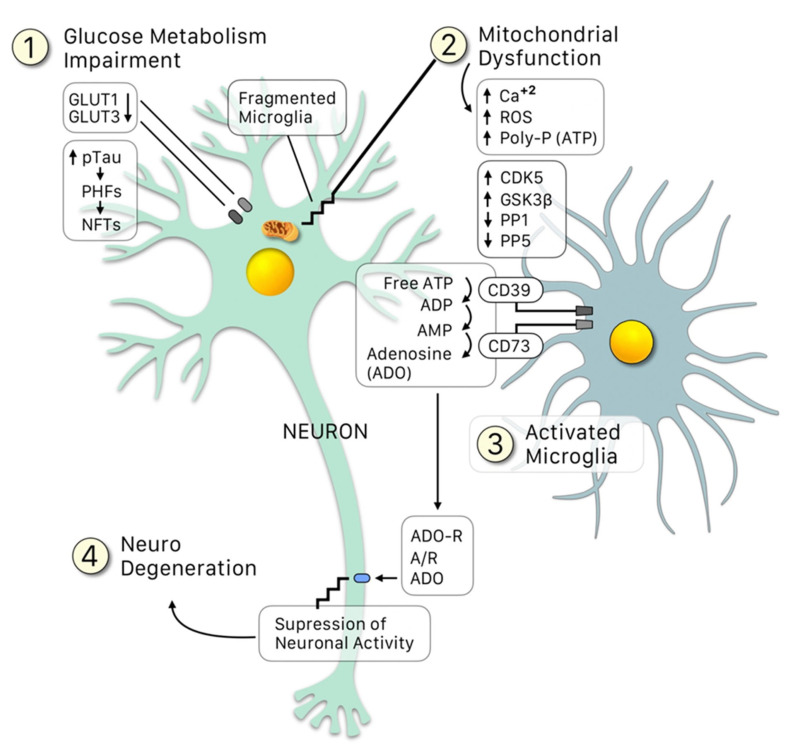
Crosstalk between neurons and microglia: An update of the Neuroimmunomodulation theory [2,15], in which besides the conventional “damage signals”, the glucose metabolism impairment leads to a downregulation of the GLUT1 and GLUT3 receptors, resulting in: (1) Loss of providing enough glucose for the mitochondria to function properly (2) All the latter leading to upregulation of the kinases (CDK5, GSK3β) and downregulation of the phosphatases (PP1 and PP5), which increase p-tau, generating PHF and NFT. Fragments of these damaged mitochondria, alongside NFT fragments (among other signals), activate the microglia (3). The microglia is the able to sense an increase in the free ATP (due to the fragmentation of the mitochondria) and transforms it into adenosine mediated by the clusters CD39 and CD73. Adenosine then binds to its receptor, ADO-R, over-suppressing the neuronal activity, thus leading to neurodegeneration.

**Figure 2 ijms-23-04192-f002:**
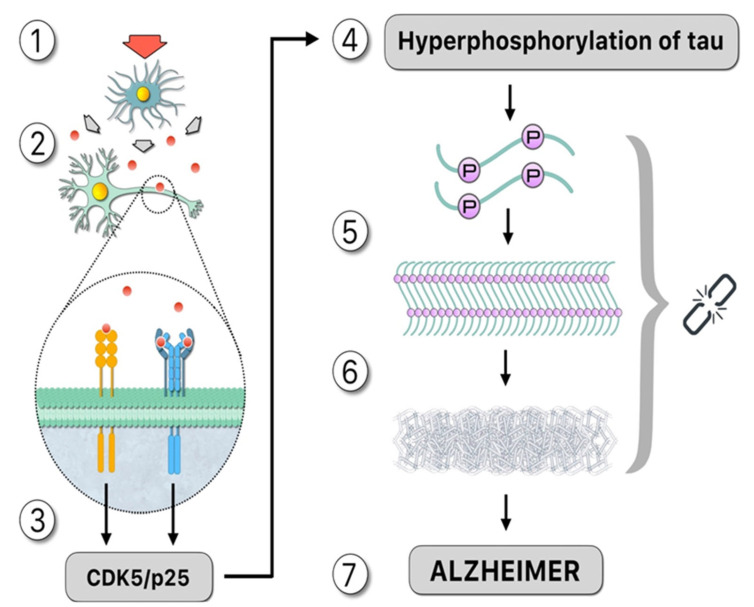
Upstream: “Damage signals”, represented by the red arrow, activate the microglia that release cytotoxic compounds such as cytokines, as demonstrated by the theory of neuroimmunomodulation [15]: (1). These proinflammatory cytokines, such as IL-1 and IL-6, signaled by red circles, activate membrane receptors in the neuron (2) which activates a signaling pathway that promotes the formation of the CDK5/p25 complex, which is mainly responsible for tau hyperphosphorylation (3). Downstream: Hyperphosphorylated tau increasing the number of binding sites for phosphates in the same molecule (4). Structural changes occur as a result of post-translational modifications, facilitating the formation of oligomers in neurons. Therefore, the ‘native’ protein is altered, and subsequently tau folds abnormally (misfolding) (5). Hyperphosphorylated tau, not bound to microtubules, binds in pairs, forming paired helical filaments (PHF). The polymerization of tau requires inductor molecules such as polyanions (6). Consequences of this pathological process include neural dysfunction or death from apoptosis. In this case, the PHF moves to the extracellular fluid accessing the nearby cell, acting as a nucleation seed and forming new aggregates of tau, and AD disease is observed (7). Symbology: 1.—Red arrow: damage signal. 2.—Red circle: cytokines. 3.—Gray key: hyperphosphorylation process. 4.—Chain: the missing link.

## Data Availability

Not applicable.

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
