# Peer review of "Alzheimer’s Disease and Tau Self-Assembly: In the Search of the Missing Link"

_ijms, 2022, doi:10.3390/ijms23084192_

Round 1
Reviewer 1 Report
This article deals with an interesting are of research; kinases, phosphates and AD. The manuscript needs few improvements before it can be worthy of publication.
- It is currently known the influence of protein kinases and phosphatases on the hyperphosphorylation of tau, and consequently, to its oligomerization. The authors are unable to explain proper meaning out of this sentence and need to be rewritten.
- Two main protein aggregates have been reported as the major role players in regard. Rewrite this.
- Introduction: Alzheimer’s pathogenesis. Adding few relevant citations will be helpful.
https://doi.org/10.1080/07391102.2019.1595728
https://doi.org/10.1007/978-981-13-9636-6_2
- It should be noted, however, that not only the hyperphosphorylation is pivotal on AD, but also which of the putative phosphorylation sites are target of the kinases. What authors are trying to convey by this sentence?
- hyperphosphorylation of tau relies in a shift of. hyperphosphorylation of tau relies on a shift of
- Several recent pieces of literature have reported the importance of MARK4 in AD via tau hyperphosphorylation. The authors are advised to add little information regarding the same under the section Hyperphosphorylation of tau: the influence of protein kinases and phosphatases during the course of AD. Adding few of these articles will be helpful in that aspect
https://doi.org/10.3390/biom10050789
https://doi.org/10.1016/j.ijbiomac.2020.06.078
https://doi.org/10.3390/ijms222010986
- we propose that ATP, after release to cytosol, may act as a upregulator/subtrate of the CDk5 kinase, which, in turn hyperphosphorylates Are authors very sure about this hypothesis. Do they have literature to back their hypothesis?
Reviewer 2 Report
This review describes tau modifications, their misfolding, initiation of its pathological selfassembly, and the role of hyperphosphorylation. This is well written and can be accepted as it is. Having said that, I believe adding a section about Tau binding sites and how phosphorylation might affect Positron emission tomography (PET) designing strategy could be interesting. Tau is known to have multiple PET binding sites (Int. J. Mol. Sci. 2021, 22(1), 349 and Acta Neuropathologica volume 141, pages697–708 (2021) and wonder if authors can comment on the importance of phosphorylation in PET designing.
